# ARINBEV: Bird's-Eye View Layout Estimation with Conditional Autoregressive Model

**Jiyong Kwag, Charles Toth, Alper Yilmaz**
The Ohio State University

## Abstract

Recent advances in Bird's Eye View (BEV) layout estimation have advanced through refinements in architectural and geometric design. However, existing methods often overlook the structured relationships among traffic elements. Components such as drivable areas, lane dividers, and pedestrian crossings constitute an interdependent system governed by civil engineering standards. For instance, stop lines precede crosswalks, which align with sidewalks, while lane dividers follow road curvature. To capture these interdependencies, we propose **ARINBEV**, an autoregressive model for BEV map estimation. Unlike prior generative approaches that rely on complex multiphase training or encoder-decoder architectures, ARINBEV employs a single-stage, decoder-only autoregressive design. This architecture enables semantically consistent BEV map estimation. On nuScenes and Argoverse2, ARINBEV attains 64.3 and 65.6 mIoU, respectively, while using $1.7\times$ fewer parameters and training $1.8\times$ faster than state-of-the-art models.

## 1 Introduction

BEV perception aligns multi-view images into a unified top-down representation to support downstream tasks in autonomous driving. Recent methods (Li et al., 2024; Liu et al., 2023b; Philion & Fidler, 2020), including depth-based and attention-based approaches, have achieved strong performance by leveraging camera parameters to fuse multi-view features. However, these methods often overlook the semantic structure of maps. Traffic elements such as drivable areas, lanes, pedestrian crossings, and sidewalks are designed manually and are governed by structured dependencies. For instance, stop lines typically precede crosswalks. These conditional dependencies are formally specified by transportation standards. For instance, MUTCD Section 3A.05 (United States Department of Transportation, 2009a) states: *"Yellow markings for longitudinal lines shall delineate the separation of traffic traveling in opposite directions and white markings for longitudinal lines shall delineate the right-hand edge of the roadway."* Similarly, Section 3B.16 (United States Department of Transportation, 2009b) specifies: *"Stop lines should be used to indicate the point behind which vehicles are required to stop in compliance with a STOP (R1-1) sign, a Stop Here For Pedestrians (R1-5b or R1-5c) sign."* Such structured relationships align with the autoregressive next-token prediction paradigm, where the presence of a traffic element can be inferred from its surrounding context. This observation motivates us to investigate conditional autoregressive modeling for BEV map estimation.

Existing generative approaches for BEV map estimation can be broadly categorized into two types: (1) two-stage frameworks (Zhang et al., 2024; Zhu et al., 2023; Zhao et al., 2024) and (2) encoder-decoder generative frameworks (Zou et al., 2024; Ji et al., 2023). Figure 1 provides an overview comparison. Following prior image generation methods, two-stage frameworks first learn a discrete representation of the ground-truth BEV map using VQ-VAE (Van Den Oord et al., 2017) reconstruction and codebook. In the second stage, a transformer (Vaswani et al., 2017; Dosovitskiy et al., 2020) trains on the learned discrete tokens to estimate the BEV map. While conceptually straightforward, this approach is not well suited for BEV map estimation. The primary purpose of discrete representation learning with VQ-VAE is to map high-dimensional continuous representations of natural images into discrete tokens. However, BEV maps are not natural images; BEV maps are highly structured, semantically constrained, and dominated by sparse backgrounds. This discrepancy often leads to codebook under-utilization, feature under-representation, and weaker supervision for the

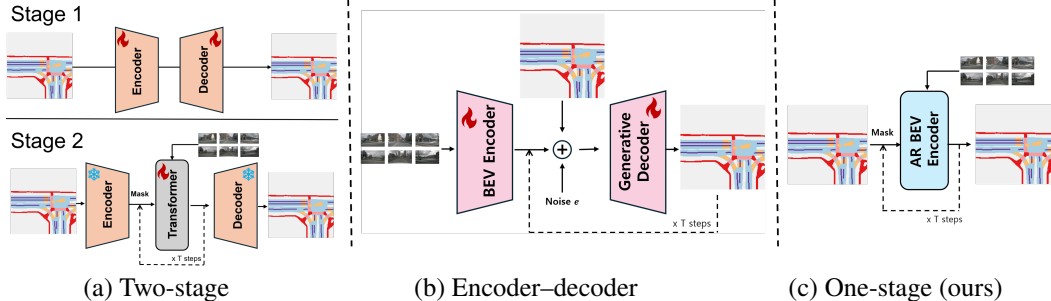

Figure 1: **Comparison of generative based BEV map estimation pipelines.** (a) Two-stage models use discrete tokenization, which increases training complexity. (b) encoder-decoder models omit discrete tokenization but still separate the BEV encoder and generative decoder stages. (c) ARIN-BEV unifies both into a single-stage, decoder-only autoregressive framework, reducing complexity while maintaining accuracy.

second-stage transformer. In contrast, encoder-decoder generative frameworks directly employ a BEV encoder to produce BEV features, which are then refined by a generative decoder. Although more unified, this approach introduces additional complexity and longer training times due to the overhead of the generative decoder. Moreover, scaling the architecture further increases model complexity quadratically, as BEV perception encoding depends on generative decoding for improved BEV map estimation. The encoder extracts features only once, limiting the advantage of iterative refinement to the decoder.

We begin with an empirical analysis of discrete representation learning using recent different quantization-based VAE (Van Den Oord et al., 2017; Mentzer et al., 2023; Yu et al., 2023) on the BEV map. Across all setups, we observe that quantization-based VAE consistently suffers from codebook under-utilization and never achieve full codebook usage (Figure 3). However, this alone does not imply that under-utilization directly causes feature under-representation, which is often cited as the primary reason for the degradation of discrete representation learning. To investigate further, we analyze the problem through Shannon entropy (Shannon, 1948). High entropy reflects greater semantic complexity, while low entropy indicates confident assignments. Our analysis shows that entropy in BEV maps is extremely low, resulting in feature under-representation except in specific central regions (Figure 2). This observation raises a key question: *Is first-stage discrete representation learning necessary for autoregressive BEV map estimation?*

To this end, we propose **ARINBEV**, a single-stage, decoder-only autoregressive model for BEV map estimation. Designing such a model presents two key challenges. The first is how to construct discrete input tokens without the first stage training (Esser et al., 2021; Sun et al., 2024; Yu et al., 2021). The second is that, while mask scheduling is essential for learning token dependencies, random mask strategies (Chang et al., 2022; Besnier et al., 2025; Yu et al., 2021) developed for natural images may be suboptimal for BEV map estimation. To address these challenges, we introduce two core components: class encoding and entropy-guided mask scheduling. Class encoding directly incorporates ground-truth BEV map labels via a lightweight embedding layer, yielding an efficient input representation. Entropy-guided masking leverages the observation that central BEV map regions contain the most uncertain and informative content, ensuring that the autoregressive model allocates capacity to learning the most critical dependencies. Together, these designs enable ARIN-BEV to perform autoregressive BEV map estimation over informative regions, eliminating the need for a separate discrete representation learning stage. We summarize the main contributions of this paper as follows:

- We demonstrate that discrete representation learning is unnecessary for generative BEV map estimation, and that encoder-decoder architectures introduce additional complexity.

- We propose **ARINBEV**, a single-stage, decoder-only autoregressive framework that removes the need for two-stage training and encoder-decoder architectures.

- Our framework achieves 64.3 and 65.6 mIoU on nuScenes and Argoverse2, respectively, setting new state-of-the-art records in BEV map estimation while using $1.7\times$ fewer parameters and achieving $1.8\times$ faster training.

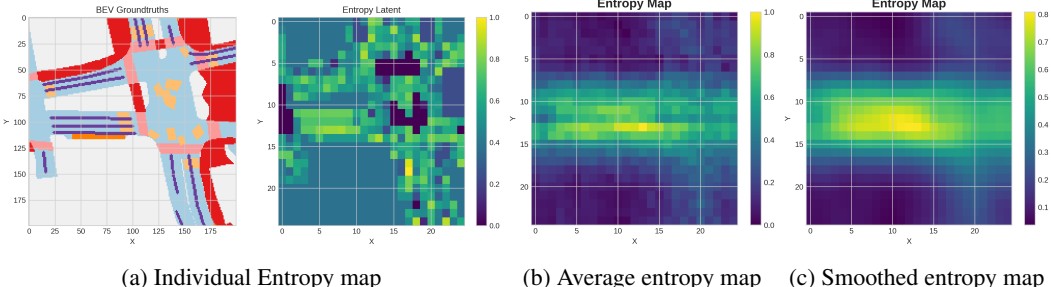

(a) Individual Entropy map      (b) Average entropy map    (c) Smoothed entropy map

Figure 2: **Quantitative results on nuScenes validation.** (a) Ground-truth BEV map with its VQ-VAE entropy map, showing local uncertainty in codebook assignments. (b) Average entropy map across the dataset, highlighting regions of higher semantic variation. (c) Smoothed average entropy map, making central informative regions more visible.

## 2 RELATED WORK

We discuss the most relevant literature and compare the major differences between traditional and generative BEV map estimation.

**Traditional BEV Map Estimation.** The most straightforward approach to BEV map estimation is the prediction of the monocular BEV map (Lu et al., 2019; Zhao et al., 2024; Gong et al., 2022; Zou et al., 2024; Roddick & Cipolla, 2020), where each camera view is processed independently. However, this setting does not capture spatial correlations across views, leading to inefficiency and limited perception. To overcome this, recent works (Philion & Fidler, 2020; Li et al., 2024; Liu et al., 2023b; Borse et al., 2023; Ge et al., 2023; Xie et al., 2022) have proposed unified frameworks that fuse multi-view images into a single BEV representation, enabling BEV map estimation on surround view. Pioneering approaches employ depth estimation (Philion & Fidler, 2020; Liu et al., 2023b; Lu et al., 2025) or attention-based mechanisms (Zhou & Krähenbühl, 2022; Li et al., 2024) to project image features into a shared top-down view, showing strong performance in BEV map estimation. Building upon these foundations, subsequent works (Li et al., 2022; Liu et al., 2023a; Liao et al., 2023) have introduced specialized decoders to convert dense BEV segmentation maps into compact vectorized representations, addressing memory efficiency.

**Generative based BEV Map Estimation.** Recent research has explored generative modeling for BEV map estimation. TaDe (Zhao et al., 2024) proposes a two-stage architecture in which a ground truth map is encoded in the first stage and then used to supervise a second-stage generator. However, TaDe is limited to monocular structures and cannot generalize to multi-view settings. MapPrior (Zhu et al., 2023) incorporates a VQGAN-style (Esser et al., 2021; Goodfellow et al., 2014) model with a discriminator to improve realism and structural detail. Although effective, it suffers from computationally expensive two-stage training and adopts encoder–decoder architecture in the second stage, increasing the cost of inference. Moreover, MapPrior relies on GPT-style 1D token generation, which is misaligned with the 2D nature of BEV maps. VQ-Map (Zhang et al., 2024) follows a similar two-stage design to MapPrior and TaDe, but supervises the second stage BEV feature using embeddings from a lightweight first stage encoder without autoregressive decoder. Although more efficient than MapPrior, its performance remains constrained by the first-stage encoder's capacity. Other generative approaches based on diffusion (Song et al., 2020; Ho et al., 2020; Ho & Salimans, 2022; Rombach et al., 2022) have also been explored. DDP (Ji et al., 2023) proposes encoder–decoder diffusion model to avoid two-stage training, but the diffusion process remains computationally heavy. DiffBEV (Zou et al., 2024) similarly employs diffusion for the BEV map estimation, but is also limited to monocular input, restricting its applicability to multi-view systems in the real world.

## 3 METHOD

Conventional two-stage models first learn discrete representations of the ground-truth BEV map via VQ-VAE (Van Den Oord et al., 2017), producing discrete tokens, which are then used for

transformer-based token prediction. We revisit this approach by analyzing codebook utilization and entropy in VQ-VAE applied to BEV map. Our findings indicate that complex two-stage and encoder-decoder pipelines are unnecessary for BEV map estimation, and that entropy maps provide valuable guidance for designing effective mask scheduling to capture conditional dependencies.

## 3.1 DISCRETE REPRESENTATION LEARNING

Unlike language, where tokens carry clear semantics, individual image pixels do not. Discrete representation learning transforms complex continuous visual data into controllable discrete tokenized forms, enabling transformer usage in vision tasks. However, BEV map labels are already sparse and discrete, resembling tokens. This raises a key question: *Is discrete representation learning necessary for autoregressive transformers when inputs are inherently token-like?*

**VQ-VAE.** We begin by reviewing discrete representation learning for BEV map using VQ-VAE. The VQ-VAE architecture consists of three main components: the encoder, the vector quantization layer, and the decoder. The encoder $\mathcal{E}(\cdot)$ is a deterministic mapping that takes an input BEV map $\mathbf{M} \in \{0,1\}^{C \times H \times W}$, where $C$ denotes the number of semantic classes, and produces a latent representation $\mathcal{E}(\mathbf{M})$. The vector quantization layer $\mathcal{Q}(\cdot)$ computes the discrete latent variable $z$ via nearest-neighbor lookup in the codebook $\mathbf{e} \in \mathbb{R}^{K \times D}$ using the encoder output $\mathcal{E}(\mathbf{M})$. Here, $K$ is the number of codebook entries, and each codebook vector is denoted $\mathbf{e}_i \in \mathbb{R}^D$ for $i = 1, \ldots, K$. Formally, the quantization is defined as:

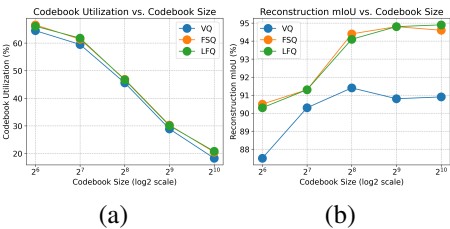

(a)        (b)

Figure 3: **Quantitative results on nuScenes validation.** (a) Utilization drops with larger codebooks. (b) Reconstruction mIoU improves with codebook size, showing no collapse.

$$\mathcal{Q}(z = i \mid \mathbf{M}) = \begin{cases} 1 & \text{if } i = \arg\min_j \|\mathcal{E}(\mathbf{M}) - \mathbf{e}_j\|_2, \\ 0 & \text{otherwise.} \end{cases} \quad (1)$$

The decoder $\mathcal{D}(\cdot)$ reconstructs the BEV map $\tilde{\mathbf{M}}$ from the quantized latent $z$. VQ-VAE training includes three losses: the reconstruction loss $\|\mathbf{M} - \tilde{\mathbf{M}}\|_2^2$, the codebook loss $\beta \|\mathrm{sg}[\mathcal{E}(\mathbf{M})] - \mathbf{e}\|_2^2$ to update codebook vectors, and the commitment loss $\|\mathcal{E}(\mathbf{M}) - \mathrm{sg}[\mathbf{e}]\|_2^2$ to encourage the encoder output to remain close to the selected codebook vector. Here, $\mathrm{sg}[\cdot]$ denotes stop-gradient, and $\beta$ is a hyperparameter, typically set in the range $[0.25, 2]$. The overall optimization objective is:

$$\mathcal{L}(\tilde{\mathbf{M}}) = \|\mathbf{M} - \tilde{\mathbf{M}}\|_2^2 + \beta \|\mathcal{E}(\mathbf{M}) - \mathrm{sg}[\mathbf{e}]\|_2^2 + \|\mathrm{sg}[\mathcal{E}(\mathbf{M})] - \mathbf{e}\|_2^2. \quad (2)$$

**Codebook Utilization.** We analyze codebook utilization in VQ-VAE trained on ground-truth BEV maps. Utilization measures the number of unique codebook entries used during validation: higher rates indicate greater representation diversity, while low rates suggest codebook collapse or limited input variation, both of which hinder transformer-based generative modeling. Figure 3 reports utilization rates for various discrete representation methods (Van Den Oord et al., 2017; Mentzer et al., 2023; Yu et al., 2023). Across all methods, utilization declines as codebook size increases, and full usage is never achieved without a drop in reconstruction mIoU, even for small codebooks. These results highlight a key insight: the inherent sparsity of BEV maps limits the semantic richness of latent representations. Large background regions contribute little diversity, leading to codebook under-utilization. This finding further suggests that discrete representation learning may be unnecessary for BEV map estimation.

**Shannon Entropy Analysis.** Low codebook utilization can arise from either true feature under-representation or codebook collapse, where only a few entries are used regardless of input diversity. To assess representational quality, we compute the Shannon entropy (Shannon, 1948) of the soft assignment probabilities between encoder features and codebook entries. Higher entropy reflects greater semantic diversity and uncertainty in the assignments, whereas lower entropy indicates more confident assignments. Hence, entropy serves as a proxy for the representational richness of BEV maps. Formally, the Shannon entropy of a discrete probability distribution $\mathbf{p} = (p_1, \ldots, p_K)$ over

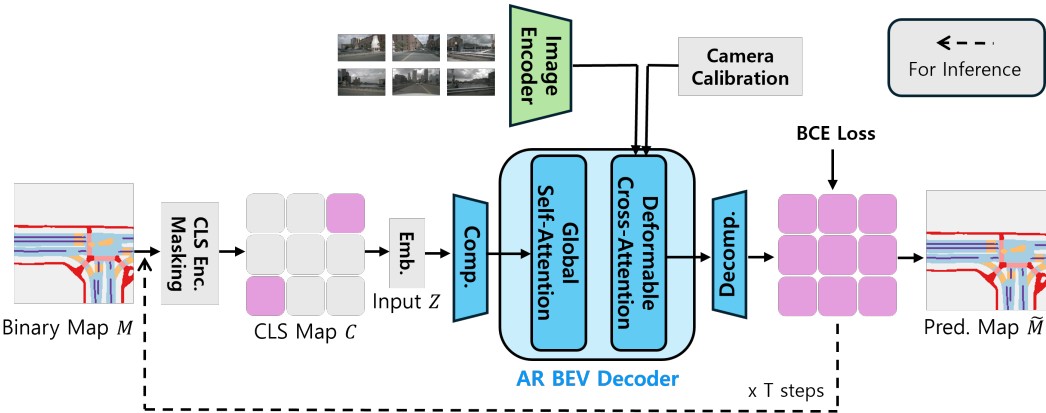

Figure 4: **Overview of the proposed ARINBEV architecture.** Multi-view images are first processed by an image encoder and projected into BEV space using camera calibration. The semantic BEV map is masked and compressed before being passed through transformer blocks. The output is subsequently decompressed to reconstruct the complete BEV map. During inference, a fully masked map is iteratively refined to estimate the BEV map.

$K$ codebook entries is defined as

$$\mathcal{H}(\mathbf{p}) = -\sum_{i=1}^{K} p_i \log p_i, \tag{3}$$

where $p_i$ denotes the probability of assigning a feature to the $i$-th codebook entry. Let $\mathbf{x} \in \mathbb{R}^{D \times H \times W}$ be the encoder feature map $\mathcal{E}(\mathbf{M})$. To compute entropy at each spatial location $(h, w)$, we define a probability distribution over the feature vector $\mathbf{x}_{h,w} \in \mathbb{R}^D$. This distribution must satisfy $p_i \geq 0$ for all $i \in \{1, \dots, K\}$ and $\sum_{i=1}^{K} p_i = 1$. Since raw feature vectors do not satisfy these constraints, we first compute the cosine similarity between the normalized feature vector $\mathbf{x}_{h,w}$ and each codebook entry $\mathbf{e}_i \in \mathbb{R}^D$ from the learned codebook $\mathbf{e} = \{\mathbf{e}_i\}_{i=1}^{K}$:

$$s_i(h, w) = \frac{\mathbf{x}_{h,w} \cdot \mathbf{e}_i}{\|\mathbf{x}_{h,w}\| \, \|\mathbf{e}_i\|} \tag{4}$$

These similarities are transformed into a valid probability distribution via the softmax function: $p_i(h, w) = \mathrm{softmax}(s_i(h, w))$. Finally, the Shannon entropy at each location $(h, w)$ is computed as

$$\mathcal{H}(h, w) = -\sum_{i=1}^{K} p_i(h, w) \log p_i(h, w). \tag{5}$$

Figure 2 illustrates the entropy map for a single BEV sample, average entropy map of the entire dataset, and its Gaussian-smoothed version. Background regions exhibit consistently low entropy, while semantically meaningful areas—such as intersections, roads, and boundaries—show higher entropy. This confirms the semantic sparsity of BEV maps, where large portions contribute little information. Consequently, forcing discrete representation learning may be unnecessary or even detrimental.

## 3.2 AUTOREGRESSIVE MODELING

In BEV maps, stage-one discrete representation learning often fails due to low-entropy regions, leading to poor codebook utilization and weak transformer (Vaswani et al., 2017) supervision. This suggests that discrete quantization may not be necessary. However, it raises a key question: *Without learned discrete tokens, how can we construct semantically meaningful index sequences for autoregressive BEV map modeling?*

**Class Encoding.** To bypass discrete representation learning and provide semantically grounded input, we begin with a binary ground-truth BEV map $\mathbf{M}$. We construct class-conditioned embeddings

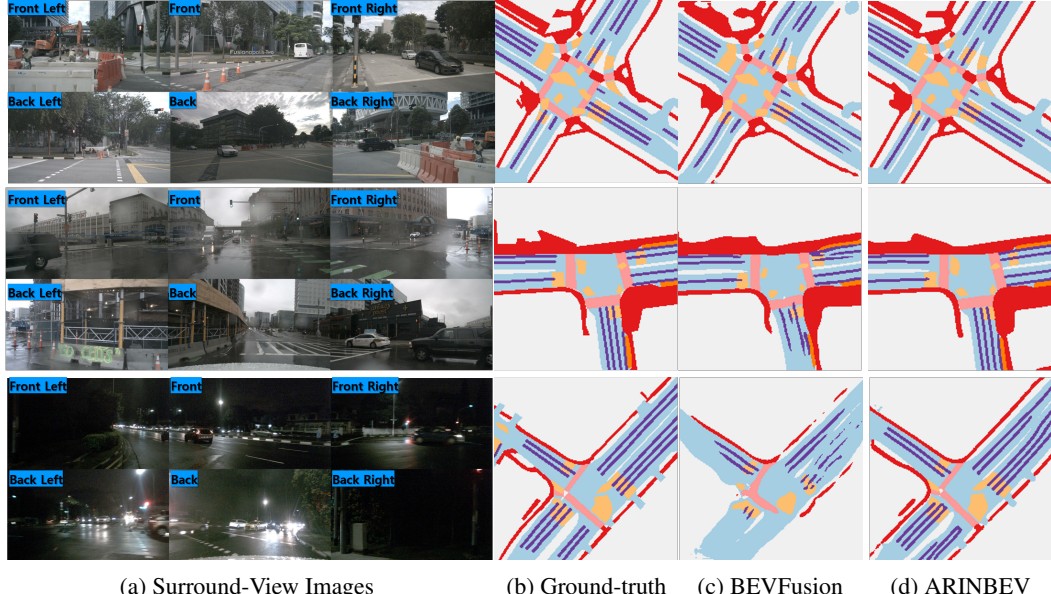

|  (a) Surround-View Images | (b) Ground-truth | (c) BEVFusion | (d) ARINBEV |

Figure 5: **Prediction results under different conditions (day, rainy, night; top to bottom) on nuScenes validation.** ARINBEV yields more plausible outputs, even in occluded regions, while reducing artifacts. Color scheme follows (Liu et al., 2023b).

using a learnable embedding table $\mathbf{E} \in \mathbb{R}^{(C+1)\times D}$, where $C$ is the number of classes and $D$ is the embedding dimension. An additional entry is reserved for the mask token. To encode the class strength, we define a multiplicative factor $\mathbf{F} \in \mathbb{R}^{1\times C\times 1\times 1}$, assigning each channel a weight equal to its class index. The weighted label map is computed as $\mathbf{C} = \mathbf{M} \odot \mathbf{F}$. The class index of each pixel in $\mathbf{C}$ is used to index in the embedding table, producing $\mathbf{S} \in \mathbb{R}^{C\times H\times W\times D}$. We then average over the class dimension:

$$\mathbf{S}_{\text{avg}} = \frac{1}{C} \sum_{i=1}^{C} \mathbf{S}_{[i,\dots]} \tag{6}$$

To normalize the embedding, we apply a bounded nonlinearity:

$$\mathbf{Z} = (2 \cdot \sigma(\mathbf{S}_{\text{avg}}) - 1) \cdot \beta \tag{7}$$

where $\sigma(\cdot)$ is the sigmoid function used for normalization, and $\beta = 0.01$ is a scaling factor for numerical stability. The resulting $\mathbf{Z}$ serves as the initial BEV feature, enabling the direct injection of semantic labels into the transformer. This formulation eliminates the need for stage-one quantization. Additional discussion is provided in the Appendix B.

**Autoregressive BEV Decoder.** The goal of our autoregressive BEV decoder is to learn parameters $\theta$ that maximize the likelihood of a token sequence $\mathbf{x} = \{x_1, x_2, \dots, x_S\}$ conditioned on multi-view image features $\mathbf{c}$ in a bidirectional manner. We decompose the token distribution according to a mask scheduling strategy $\mathcal{S}$:

$$p(\mathbf{x} \mid \mathbf{c}) = \prod_{s=1}^{S} p(\mathbf{x}_s \mid \mathbf{x}_{<s}, \mathbf{c}), \tag{8}$$

where $\mathbf{x}_s$ denotes the set of tokens predicted at step $s$ according to the schedule $\mathcal{S}$, and $\mathbf{x}_{<s}$ represents the tokens already predicted. Our architecture builds upon BEVFormer (Li et al., 2024) and DETR (Carion et al., 2020), using pre-normalized attention layers (Dosovitskiy et al., 2020). The input embedding $\mathbf{Z}$ is initialized from ground-truth BEV map labels via a learnable embedding table $\mathbf{E}$. To reduce the cost of deformable cross-attention (Zhu et al., 2020), we compress $\mathbf{Z}$ from $200 \times 200$ to $25 \times 25$, followed by a decompression layer to restore resolution. Given the sparsity of compressed tokens, deformable self-attention is replaced with global self-attention. For cross-view interaction, we retain deformable cross-attention to access relevant image features using learned offsets and camera parameters. Binary focal loss (Lin et al., 2017) is applied to mitigate class imbalance in per-token classification. Figure 4 provides an overview of the proposed architecture.

Table 1: **State-of-the-art comparison for surround-view BEV map estimation on the nuScenes validation set.** All methods are evaluated under the camera-only setting. The results include per-class IoU (%) scores for six semantic categories and the mean IoU.

| Methods | IoU ↑ (%) | | | | | | |
|---|---|---|---|---|---|---|---|
| | Drivable | Ped. Cross. | Walkway | Stopline | Carpark | Divider | Mean |
| OFT | 74.0 | 35.3 | 45.9 | 27.5 | 35.9 | 33.9 | 42.1 |
| LSS | 75.4 | 38.8 | 46.3 | 30.3 | 39.1 | 36.5 | 44.4 |
| CVT | 74.3 | 36.8 | 39.9 | 25.8 | 35.0 | 29.4 | 40.2 |
| M$^2$BEV | 77.2 | – | – | – | – | 40.5 | – |
| BEVFusion | 81.7 | 54.8 | 58.4 | 47.4 | 50.7 | 46.4 | 56.6 |
| MapPrior | 81.7 | 54.6 | 58.3 | 46.7 | 53.3 | 45.1 | 56.7 |
| X-Align | 82.4 | 55.6 | 59.3 | 49.6 | 53.8 | 47.4 | 58.0 |
| MetaBEV | 83.3 | 56.7 | 61.4 | 50.8 | 55.5 | 48.0 | 59.3 |
| DDP | 83.6 | 58.3 | 61.6 | 52.4 | 51.4 | 49.2 | 59.4 |
| VQ-Map | 83.8 | 60.9 | 64.2 | 57.7 | 55.7 | 50.8 | 62.2 |
| **ARINBEV** | **85.0** | **62.4** | **66.5** | **60.8** | **59.7** | **51.2** | **64.3** |

**Entropy-Guided Masking Strategy.** In BEV map estimation, Figure 2 shows that semantic information is concentrated near the center, while peripheral regions are mostly background. To guide learning toward informative areas, we adopt a spatial masking strategy based on a Halton sequence (Halton, 1964), inspired by Halton MaskGIT (Besnier et al., 2025) with entropy-guided selection. To mitigate overfitting to dataset-specific entropy, we employ a fixed Gaussian prior map $\mathbf{S} \in \mathbb{R}^{H \times W}$, centered on the grid with standard deviation $\sigma = 0.5$. The masking ratio is obtained by sampling $r \sim \mathcal{U}(0, 1)$ and transforming it via the arccos distribution:

$$\rho_b = \frac{2}{\pi} \arccos(r). \tag{9}$$

Next, we generate $N$ candidate coordinates $\{(y_k, x_k)\}_{k=1}^N$ using a random Halton sequence and assign selection probabilities proportional to the Gaussian prior map $\mathbf{S}$:

$$p_k = \frac{S_{y_k, x_k}}{\sum_{j=1}^N S_{y_j, x_j}}. \tag{10}$$

We then sample $\lfloor \rho_b \cdot H \cdot W \rfloor$ positions without replacement according to $p_k$ to construct a mask $\mathbf{B}$. To further prevent overfitting to sparse unmasked regions, this stochastically guided entropy masking is applied with probability $p = 0.5$, combined with random arccosine masking as in MaskGIT.

## 4 EXPERIMENTS

Table 2: **Computational overhead analysis.** Training time is measured in GPU hours using a single NVIDIA A100 on the nuScenes training set. ARINBEV achieves the shortest training time compared to prior work, requires fewer parameters than two-stage models VQ-Map and MapPrior, and incurs lower computational cost (MACs) than encoder–decoder model DDP.

| Methods | Params (M) | MACs (G) | Train Time (h) | mIoU ↑ (%) |
|---|---|---|---|---|
| BEVFusion | 50.1 | 155.5 | 100 | 56.6 |
| MapPrior | 719.1 | 396.0 | > 200 | 56.7 |
| DDP | 53.6 | 614.1 | 160 | 59.4 |
| VQ-Map | 108.3 | 231.6 | 131 | 62.2 |
| **ARINBEV** | 63.4 | 215.8 | 73 | 64.3 |

### 4.1 EXPERIMENTAL SETTINGS

**Dataset.** We evaluate ARINBEV on BEV map estimation from surround-view images using two large-scale public datasets: nuScenes (Caesar et al., 2020) and Argoverse2 (Wilson et al., 2023).

Table 3: **Comparison of surround-view BEV map estimation on the Argoverse2 validation set.** All methods are evaluated under the camera-only setting. A constant IoU threshold of 0.5 is used across all methods to ensure fair comparison. Results marked with * are reproduced from their respective official implementations.

| Methods | IoU ↑ (%) | | | |
| --- | --- | --- | --- | --- |
| | Drivable | Ped. Cross. | Divider | Mean |
| MapPrior* | 78.5 | 50.0 | 41.1 | 56.5 |
| BEVFusion* | 80.6 | 53.1 | 47.0 | 60.2 |
| VQ-Map* | 79.7 | 56.9 | 46.9 | 61.2 |
| DDP* | 83.5 | 58.1 | 48.8 | 63.5 |
| **ARINBEV** | **83.9** | **61.4** | **51.6** | **65.6** |

nuScenes provides RGB images from six cameras covering 360° views across 1000 scenes in Boston and Singapore, with 28K frames for training and 6K for validation. Argoverse2 offers seven-camera 360° views from six US cities, comprising 110K training and 24K validation frames.

**Evaluation Protocol.** We follow the standard evaluation protocol from prior work (Liu et al., 2023b; Zhang et al., 2024; Ji et al., 2023). The perception range is set to $[-50.0\,\text{m}, 50.0\,\text{m}]$ along both the X- and Y-axes, with a segmentation resolution of 0.5 meters per pixel, resulting in a $200 \times 200$ BEV grid. Map construction quality is assessed using Intersection-over-Union (IoU) with a threshold of 0.5. For nuScenes, six binary segmentation categories are evaluated: drivable area, lane, pedestrian crossing, stop line, sidewalk, and parking area. For Argoverse2, three binary segmentation categories are used: drivable area, lane, and pedestrian crossing.

**Implementation Details.** We adopt SwinT-Tiny Liu et al. (2021) as the image backbone for a fair comparison with the prior work (Liu et al., 2023b; Zhu et al., 2023; Ji et al., 2023). We employ the AdamW optimizer (Loshchilov & Hutter, 2017) with a weight decay of 0.01. The model is trained for 20 epochs, including 8 warm-up epochs, on 4 NVIDIA A100 GPUs with a batch size of 8 per GPU. The initial learning rate is set to $5 \times 10^{-5}$ and scheduled using the one-cycle policy (Smith, 2017). During inference, we apply 3 sampling steps with the Halton scheduler(Besnier et al., 2025). Additional training details are provided in the Appendix A.

**Results.** We report BEV map estimation results on nuScenes and Argoverse2 in Table 1 and Table 3, and computation results in Table 2. Figure 5 presents qualitative visualization comparisons. On nuScenes, ARINBEV's simplified architecture—motivated by insights from extensive analysis— achieves the fastest training time among prior works. It also achieves a substantially lower parameter count compared to two-stage models such as MapPrior and VQ-Map (Zhu et al., 2023; Zhang et al., 2024). By unifying the BEV encoder and generative decoder into a single-stage model, ARINBEV further reduces the number of multiply–accumulate operations (MACs) relative to encoder–decoder baselines (Ji et al., 2023), while maintaining high efficiency. Despite its simplicity, ARINBEV outperforms more complex architectures across all map element categories. On Argoverse2, both MapPrior and VQ-Map exhibit substantial performance degradation. These models rely heavily on discrete representation learning, which is less effective when limited to the three map element classes available in Argoverse2. In contrast, ARINBEV does not depend on two-stage quantization and remains robust, achieving strong performance without suffering from this degradation.

## 4.2 ABLATION AND ANALYSIS

To investigate the contribution of individual components, we conduct an ablation study on the nuScenes validation set, as shown in Table 4.

**Masking Strategy.** To evaluate the impact of entropy-guided masking, we compare three variants: (a) *arccosine random masking*, (b) *pure entropy-guided masking*, and (c) *hybrid masking*, which switches between random and entropy-guided strategies with probability $p = 0.5$. The hybrid method (c) achieves the highest performance, outperforming both (a) and (b). This suggests that integrating entropy awareness improves model performance. All experiments use (g) as the default feature compression method.

Table 4: **Ablation study on the nuScenes validation set evaluating the impact of different modules.** Rows (a)–(c) compare various masking strategies, while (d)–(g) investigate different feature compression techniques. Row (h) evaluates an alternative embedding formulation. The best-performing and final configuration is marked with *.

| | Drivable | Ped. Cross. | Walkway | Stopline | Carpark | Divider | Mean |
|---|---|---|---|---|---|---|---|
| | | | | IoU ↑ (%) | | | |
| (a) | 83.6 | 61.5 | 64.5 | 59.1 | 58.3 | 50.1 | 62.9 |
| (b) | 82.0 | 59.4 | 62.5 | 57.8 | 56.7 | 48.1 | 61.1 |
| (c)* | 85.0 | 62.4 | 66.5 | 60.8 | 59.7 | 51.2 | 64.3 |
| (d) | 84.5 | 62.0 | 66.3 | 60.3 | 59.2 | 51.0 | 63.9 |
| (e) | 84.6 | 62.1 | 66.4 | 60.5 | 59.4 | 51.1 | 64.0 |
| (f) | 83.0 | 60.4 | 65.5 | 59.3 | 58.3 | 50.0 | 62.8 |
| (g)* | 85.0 | 62.4 | 66.5 | 60.8 | 59.7 | 51.2 | 64.3 |
| (h) | 83.0 | 59.1 | 61.2 | 58.3 | 56.6 | 47.5 | 61.0 |

**Convolution vs. Interpolation.** To assess the impact of different feature compression strategies, we fix the BEV embedding shape to $25 \times 25$ and compare the following configurations: (d) three sequential $3 \times 3$ convolutional layers, (e) a single $8 \times 8$ convolutional layer with stride 8 and zero padding, (f) pure bilinear downsampling by a factor of 8, and (g) a hybrid approach combining bilinear downsampling followed by a $3 \times 3$ convolution. Among these, (g) yields the highest mIoU, slightly outperforming the convolution-based variants (d) and (e), and significantly surpassing the pure interpolation (f). These results highlight the effectiveness of combining smooth interpolation with learnable convolution for producing compact BEV embeddings. For all experiments in this group, we use the masking strategy (c) and apply bilinear upsampling followed by a $3 \times 3$ convolution as the default decompression layer.

**Different Class Encoding.** Beyond our proposed structured class encoding, we evaluated an alternative prototype: (h) *channel-wise binary encoding*. In this formulation, each spatial location on the binary BEV map $\mathbf{M}$ is interpreted as a binary vector across channels and converted into a integer index via a dot product with a weight vector of two power $[2^{C-1}, \dots, 2^0]$. Although this approach is simpler, it produces a lower mean IoU of 61.0%. For fair comparison, we use the same default configuration with masking strategy (c) and feature compression method (g).

## 5 ACKNOWLEDGMENTS

This research/project is supported in part by the U.S. Department of Transportation under Grant 69A3552348327 for the CARMEN+ University Transportation Center.

## 6 CONCLUSION

In this work, we introduced **ARINBEV**, a simplified autoregressive BEV map estimation framework. Through codebook utilization and entropy analysis, we show that discrete representation learning in BEV semantics exhibits sparse, low-entropy structure, suggesting substantial redundancy in discrete token modeling. Through this observation, we design a single-stage, decoder-only autoregressive model with a lightweight class encoding mechanism and an entropy-guided masking schedule to focus capacity on informative regions. As a result, ARINBEV achieves state-of-the-art results on both the nuScenes and Argoverse2 benchmarks with a compact architecture and low training time. We hope our work can inspire future research on efficient BEV map estimation training.

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

## APPENDIX

## A  IMPLEMENTATION DETAILS

**Architecture Detail.** We mainly follow archtecture design from BEVFormer and DETR. However, we made several modification on architecture. Our autoregressive BEV decoder contains 8 layers with dimension of 512 and 8 attention heads using pre-normalization. BEV embeddings are downsampled by $1/8$ and upsampled by 8× using bilinear interpolation, each followed by a $3 \times 3$ convolution to enhance local context. Global self-attention is applied on the compressed grid of size $25 \times 25$. For deformable cross-attention, we sample 2 anchors from 4 heights, 2 depths, and 2 width, each predicting 2 learned offsets per scale. Figure 6 illustsrate specific deisgn of our architecture.

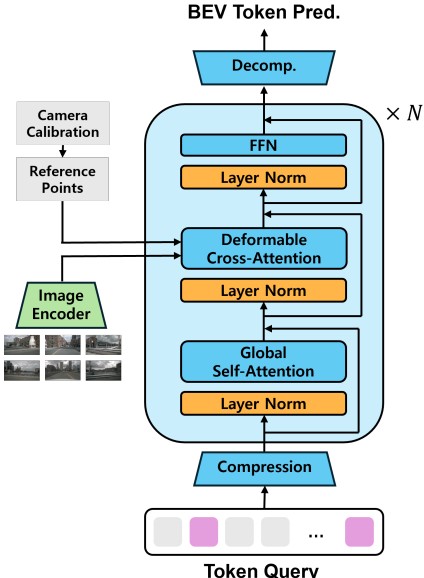

Figure 6: **More detail overview of autoregressive BEV deocder.** Our model is consisted of global self-attention, deformable cross-attention, and feedforward network with pre-normalization.

**Train Detail.** For nuScenes and Argoverse2, we use distinct pipeline as each dataset has distinct multi-view camera setups and images. For nuScenes, the six images are resized from $1600 \times 900$ to $256 \times 705$ and follow same augmentation pipeline from BEVFusion for a fair comparisonss with the prior work (Liu et al., 2023b; Ji et al., 2023; Zhang et al., 2024). For Argoverse2, the front view images $1550 \times 2048$ and other six view images $2048 \times 1550$ are padded to $2048 \times 2048$, then uniformly resized by a factor of 0.3. Data augmentation follows the MapTR pipeline Liao et al. (2023) to accommodate the different setup of Argoverse2.

**Discrete Representation Learning Detail.** For discrete representation learning, we adopt the VQ-Map (Zhang et al., 2024) encoder. The $200 \times 200$ BEV map is divided into $8 \times 8$ patches, yielding an input of shape $625 \times 8 \times 8$. These are passed through three $3 \times 3$ convolutional layers with 64, 128, and 128 channels, followed by a $2 \times 2$ max pooling layer. A linear projection reduces the channel dimension to 16 to improve codebook utilization Yu et al. (2021). Vector quantization uses exponential moving average (EMA) updates with a decay rate of 0.99 and $\epsilon = 10^{-5}$ for stability with codebook size of 512. The decoder, following VQGAN Esser et al. (2021), consists of five convolutional layers and three residual blocks.

## B  MORE EXPERIMENTS

**Class Encoding.** Class encoding allows our model to bypass the need for discrete representation learning, thereby reducing both training time and overall model complexity. As shown in Eq. 7, appropriate scaling is crucial, since different choices directly affect the stability of training. Table 5a reports results for various scaling factors: a scale of 0.01 exhibits the best performance, whereas larger values degrade accuracy. We also evaluated a learnable scale, but it consistently produced the lowest performance among all settings.

**Entropy-Guided Masking Strategy.** In entropy-guided masking, selecting an appropriate $\sigma$ for the Gaussian prior map is crucial to ensure stable training. Table 5b reports results for different $\sigma$ values. The black square represents a masked token. A setting of $\sigma = 0.3$ leads to training failure, whereas performance improves from $\sigma = 0.4$ onward. The best performance is observed at $\sigma = 0.5$, with only a marginal decrease thereafter. To evaluate the diversity of masking distributions, Table 6a reports the dynamic changes in the $\sigma$ values during training. Moreover, Figure 7 visualizes entropy-guided masking compared with random masking at the same masking ratio using $\sigma = 0.5$. We also provide a PyTorch implementation of our masking strategy in F.

**Sampling Drift.** A common explanation of declining performance after step 3 is sampling drift, where the distribution during inference differs from training. During training, the model learns to invert a masked ground-truth BEV map, whereas during inference it must iteratively unmask its own imperfect predictions under a fully masked input. As these errors accumulate, the input distribution progressively diverges from the ground-truth distributions, typically after step 3. To reduce sampling drift, we adopt the same training strategy used in DDP. We construct $Z_{model}$ from the model's predictions using unmasked inputs $Z$, and then apply masking to $Z_{model}$ during the final 9K iterations (1 epoch). However, accuracy consistently stabilizes around step 3, and additional steps do not yield further improvement in Table 6b.

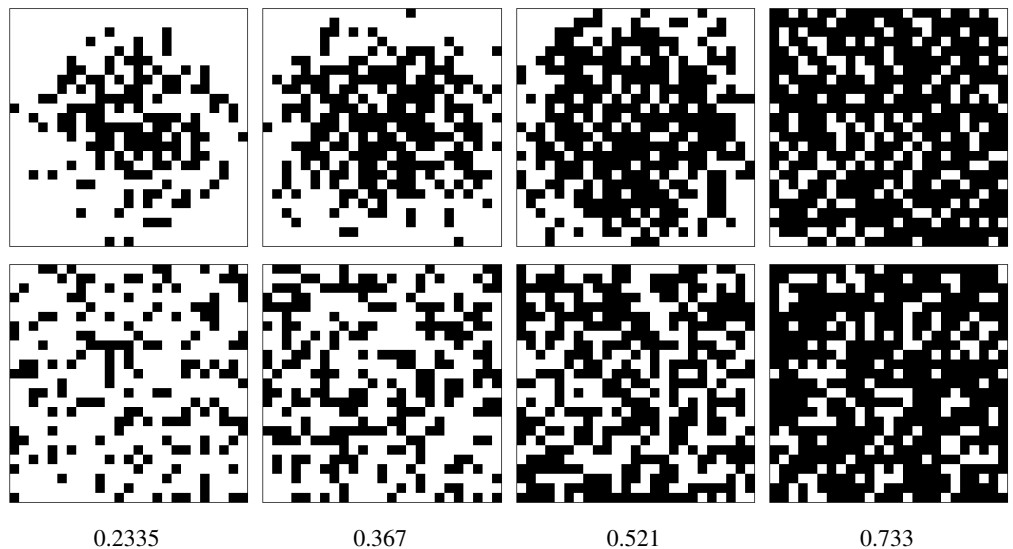

|  0.2335  |  0.367  |  0.521  |  0.733  |

Figure 7: **Comparison of entropy-guided masking with $\sigma = 0.5$ (top) and random masking (bottom) at randomly chosen masking ratios of 0.2335, 0.367, 0.521, and 0.733.** The masking is primarily concentrated in the central region, where the most informative areas reside.

Table 5: **Comparisons of scaling factors in class encoding and $\sigma$ values in entropy-guided masking.**

| Scale | mIoU |
|---|---|
| 0.01 | 64.3 |
| 0.1 | 63.2 |
| 1.0 | 61.2 |
| learned | 57.2 |

(a) Comparison of different scales on class encoding.

| $\sigma$ | mIoU |
|---|---|
| 0.3 | 62.1 |
| 0.4 | 63.8 |
| 0.5 | 64.3 |
| 0.6 | 64.0 |

(b) Comparison of different $\sigma$ values on entropy-guided masking.

Table 6: **Comparison of $\sigma$ ranges for entropy-guided masking and the impact of iterative sampling steps.**

| $\sigma$ | mIoU |
|---|---|
| 0.25-0.75 | 63.9 |
| 0.30-0.60 | 64.2 |
| 0.40-0.70 | 64.3 |

(a) Effect of different $\sigma$ ranges on masking performance.

| step | mIoU |
|---|---|
| 1 | 59.8 |
| 2 | 63.4 |
| 3 | 64.3 |
| 4 | 64.3 |

(b) Effect of iterative sampling steps.

## C  MORE VISUALIZATIONS

In this section, we provide additional visualizations on the nuScenes validation set. Figure 8 presents autoregressive predictions of the BEV map, demonstrating how the model progressively refines its estimates based on previous predictions with our efficient masking strategy. Figure 9 illustrates further examples of entropy maps on the validation set. As previously observed, background regions exhibit the lowest entropy. Moreover, certain inner regions of the road, where only drivable areas exist, also show low entropy or a consistent codebook assignment. This further strengthens our claim that discrete representation learning is unnecessary. Figures 10, 11 show additional visualizations across diverse scenarios (daytime, rainy conditions, and nighttime) on the nuScenes validation. We consistently observe that ARINBEV exhibits strong robustness despite occlusion and nighttime conditions, owing to dependency learning that enables better prediction using only observed traffic elements.

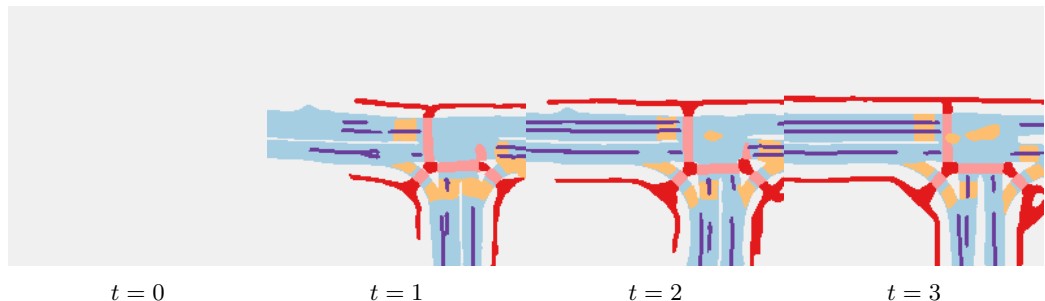

$t = 0$         $t = 1$         $t = 2$         $t = 3$

Figure 8: **Autoregressive BEV layout predictions over steps.** The model progressively refines structure and semantics.

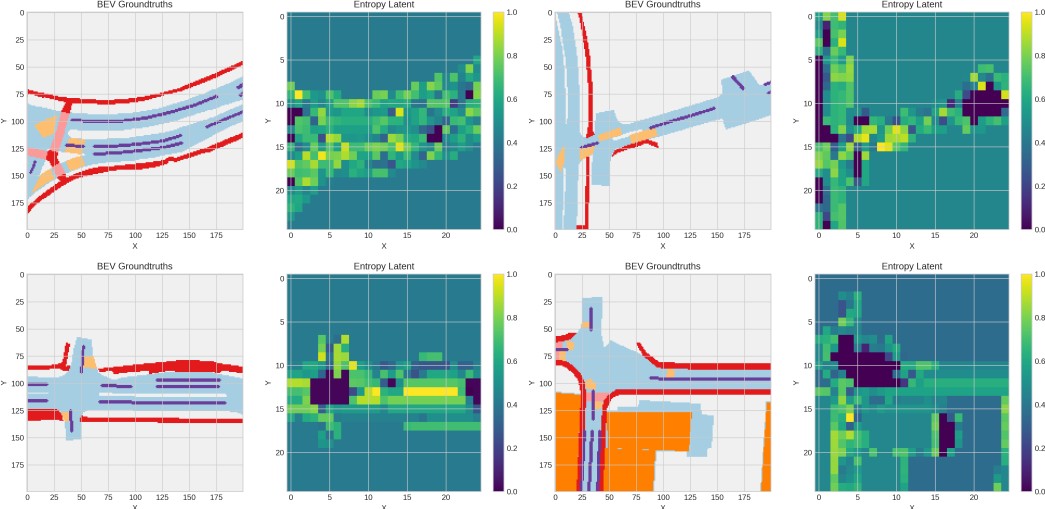

Figure 9: **More quantitative results on nuScenes validation.** Ground-truth BEV map with its VQ-VAE entropy map, showing local uncertainty in codebook assignments.

## D  LIMITATIONS AND FUTURE WORK

**Inference Speed.** We evaluate the effect of varying the autoregressive inference steps. Figure **??** shows how performance changes with different step counts. mIoU rises sharply from step 1 to step 3, reaching 64.3%, but gradually declines thereafter. These results suggest that early autoregressive refinements capture essential structural dependencies, whereas excessive iterations introduce overfitting, degrading prediction quality. FPS is highest at the beginning (18 FPS), approaching real-time

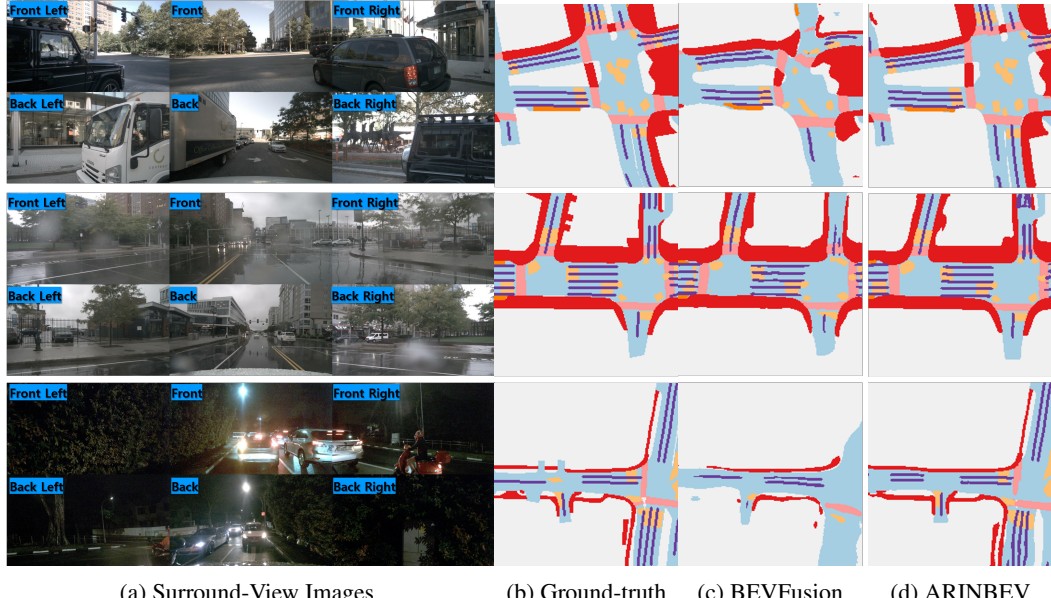

|  |  |  |  |
|---|---|---|---|
| (a) Surround-View Images | (b) Ground-truth | (c) BEVFusion | (d) ARINBEV |

Figure 10: **More Prediction results under different conditions (day, rainy, night; top to bottom) on nuScenes validation.**

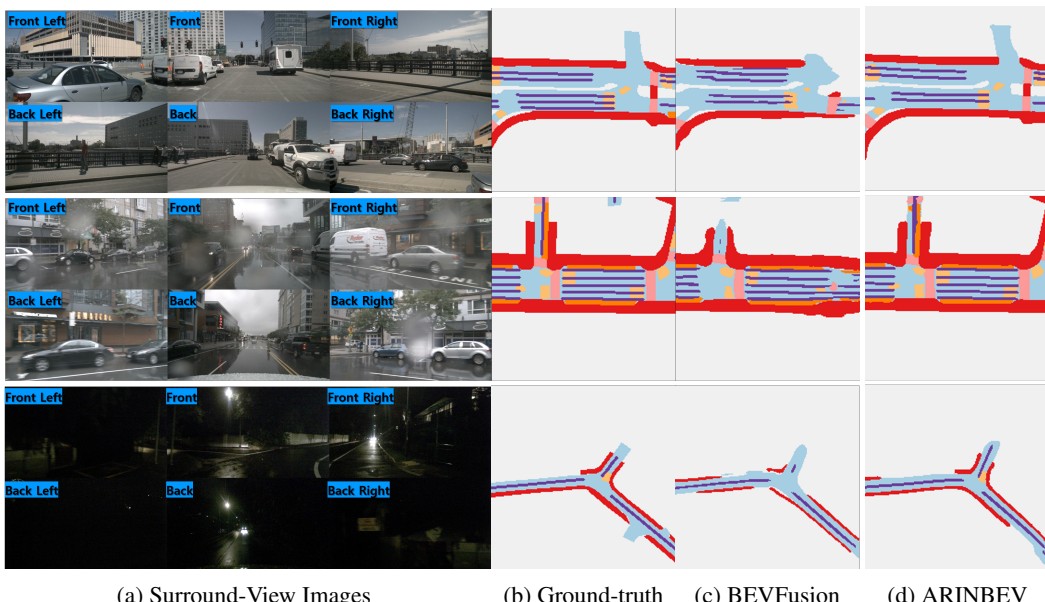

|  |  |  |  |
|---|---|---|---|
| (a) Surround-View Images | (b) Ground-truth | (c) BEVFusion | (d) ARINBEV |

Figure 11: **More prediction results under different conditions (day, rainy, night; top to bottom) on the nuScenes validation set.** We provide more challenging cases, where cameras are heavily occluded by vehicles and rain, and in night-time scenes with minimal lighting.

performance, and decreases as the number of steps increases. The best mIoU corresponds to 7.5 FPS.

**3D Occupancy Prediction.** The current work focuses on BEV map estimation in 2D space, but the dependency-based prediction scheme underlying ARINBEV provides a natural foundation for extension to 3D occupancy prediction, where modeling height variations is essential. Nonetheless, applying autoregressive modeling directly to dense 3D voxel grids may impose prohibitive computational costs. Future work could leverage ARINBEV's efficiency in 2D and investigate more scalable

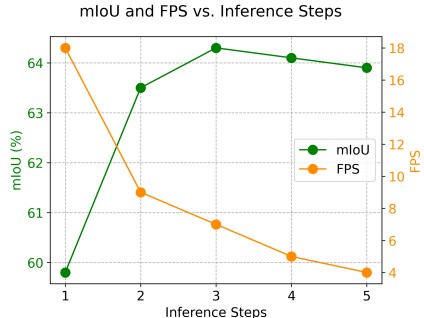

Figure 12: **mIoU and FPS vs. inference steps.** mIoU peaks at early steps and diminishes thereafter.

strategies, such as sparse voxelization, hierarchical decoding, or hybrid 2D–3D representations, to enable practical autoregressive occupancy prediction.

**Discrete Diffusion.** While ARINBEV employs discrete mask tokens for autoregressive BEV map estimation, an alternative direction is to investigate discrete diffusion with continuous noise. Diffusion models operate over the entire representation space while gradually introducing scheduled Gaussian noise, which may provide stronger generative capacity than masked-token strategies. Exploring this paradigm could offer a principled way to combine the advantages of diffusion modeling with dependency-based prediction schemes.

## E  LLM USAGE

Our primary use of LLMs is to refine our writing and receive recommendations on word choices. We did not use LLMs for retrieval or discovery of ideas, such as finding related work, nor did we use them for research ideation. Instead, all conceptual development, methodological design, and experimental analysis were conducted independently by the authors. The role of LLMs was limited to improving clarity, grammar, and presentation without influencing the scientific contributions of this work.

## F  PYTORCH IMPLEMENTATION OF ENTROPY-GUIDED MASKING STRATEGY

```python
import torch
from scipy.stats import qmc

def get_halton_coords(H, W, n_points):
    # Generate random halton sequence
    sampler = qmc.Halton(d=2, scramble=True)
    sample = sampler.random(n=n_points)
    coords = torch.tensor(sample) * torch.tensor([H, W])
    coords = torch.floor(coords).long()

    # Remove duplicates
    coords = torch.unique(coords, dim=0)
    ys, xs = coords[:, 0], coords[:, 1]
    return ys, xs

def entropy_halton_mask(gaussian_prior, code, mask_id, n_points):
    B, C, H, W = code.shape
    N = H * W

    # arccos scheduler
    r = torch.rand(B)
    val_to_mask = torch.arccos(r.clamp(0, 1)) / (math.pi * 0.5)

    masked_code = code.clone()
    mask = torch.zeros((B, H, W), dtype=torch.bool)
```

```
26
27     for b in range(B):
28         # Get Halton candidate points
29         ys, xs = get_halton_coords(H, W, n_points)
30
31         # Compute entropy probabilities
32         scores = gaussian_prior[ys, xs]
33         prob = scores / (scores.sum() + 1e-8)
34
35         # Get number of tokens to mask
36         num_tokens = int(val_to_mask[b].item() * N)
37         num_tokens = min(num_tokens, len(prob))
38
39         # Sample mask location based on probablities
40         idx_sel = torch.multinomial(prob, num_tokens, replacement=False)
41         ys_sel = ys[idx_sel]
42         xs_sel = xs[idx_sel]
43
44         mask[b, ys_sel, xs_sel] = True
45
46     mask = mask.unsqueeze(1).expand(B, C, H, W)
47
48     # Apply mask
49     masked_code[mask] = mask_id
50
51     return masked_code, mask
```

