# OpenReview forum: "ARINBEV: Bird's-Eye View Layout Estimation with Conditional Autoregressive Model"
_ICLR.cc/2026/Conference — ICLR 2026 Poster_

### Official Review · Reviewer_FeCg · 2025-10-25

**Soundness:** 3
**Presentation:** 3
**Contribution:** 4
**Rating:** 6
**Confidence:** 4

**Summary:**

This paper addresses the oversight of structured relationships among traffic elements in existing BEV layout estimation methods by proposing ARINBEV, a single-stage, decoder-only autoregressive model. Unlike two-stage or encoder-decoder frameworks, ARINBEV leverages class encoding and entropy-guided mask scheduling to bypass discrete representation learning. Evaluated on nuScenes and Argoverse2 datasets, it achieves state-of-the-art mIoU scores (64.3 and 65.6 respectively) with 1.7× fewer parameters and 1.8× faster training compared to existing models, demonstrating effectiveness in capturing semantic dependencies of traffic elements.

**Strengths:**

1. The core idea of exploiting conditional autoregressive modeling to capture structured dependencies among traffic elements (e.g., stop lines preceding crosswalks) aligns well with real-world transportation standards, bringing valuable domain awareness to BEV map estimation.
2. The single-stage, decoder-only architecture eliminates the inefficiencies of discrete representation learning and separate encoder-decoder stages, achieving superior performance with fewer parameters and faster training, which is practically meaningful for autonomous driving applications.
3. Comprehensive experiments on two large-scale datasets (nuScenes and Argoverse2) and detailed ablation studies (on masking strategies, feature compression, etc.) effectively validate the contribution of each component in the proposed framework.

**Weaknesses:**

1. The model’s advantage in this paper seems to lie in utilizing historical information via autoregressive modeling, but the authors fail to clarify whether baseline models use the same length of historical data or are video-based. This ambiguity makes it hard to attribute performance gains solely to the autoregressive design.
2. While the model achieves better results with fewer parameters and lower computation, the paper lacks in-depth insights explaining why the autoregressive paradigm inherently outperforms two-stage or encoder-decoder architectures in efficiency-performance trade-off.
3. Inference speed is critical for autonomous driving, yet the paper only reports training time and MACs. It does not compare end-to-end inference overhead (e.g., latency) across methods, which limits the evaluation of its practical applicability.
4. The ablation study on feature compression and masking strategies, though detailed, does not sufficiently discuss how these components interact with the autoregressive core, leaving gaps in understanding the model’s holistic working mechanism.

**Questions:**

1. Why is BEVFormer not included in the comparative experiments? Including it would better contextualize ARINBEV’s performance in the existing landscape.
2. For different ego trajectory types (e.g., straight, left turn, right turn, high-speed driving, start-stop), does the model’s performance improvement vary? If so, what factors contribute to these differences?
3. How does the Gaussian prior’s standard deviation (σ=0.5) in the entropy-guided masking strategy generalize to other datasets or scenarios beyond nuScenes and Argoverse2?
4. The paper mentions that excessive autoregressive inference steps lead to performance degradation. Is there a theoretical or empirical explanation for this phenomenon (e.g., overfitting to local dependencies)?

---

> ### Author Response · Authors · 2025-11-20
> **Rebuttal by Authors**
>
> Dear Reviewer FeCg,
>
> We truly appreciate your insightful review and valuable feedback. We are grateful that you highlighted the key strengths of our approach, including **(1) the use of autoregressive modeling, (2) the efficiency and simplicity of the framework,** and **(3) the comprehensive experiments and ablations.** We appreciate your constructive comments and outline our responses as follows:
>
> [Q1] **Historical information used in the autoregressive modeling is not provided**
>
> [A1] BEVFormer incorporates temporal deformable self-attention to aggregate historical BEV information. However, the aim of our autoregressive design is to model spatial structural dependencies among traffic elements through an autoregressive formulation. For this reason, we intentionally do not incorporate temporal BEV features into ARINBEV to evaluate the benefit of autoregressive spatial reasoning itself.
>
> [Q2] **Why the autoregressive paradigm inherently outperforms two-stage or encoder–decoder architectures**
>
> [A2] We thank the reviewer for highlighting the need to further clarification on our architecture advantage. For two-stage architectures, the first-stage discrete representation learning aims to map high-dimensional visual inputs into a learned discrete codebook, which is then used as the ground-truth input for a second-stage transformer. However, our empirical analysis shows that the discrete tokens suffer from under-representation (Fig. 2), a symptom of quantization-induced information loss that can weaken transformer supervision [1, 2]. Since BEV maps are already sparse and inherently discrete, applying an additional quantization stage forces the model to re-discretize an already discrete space. However, our class encoding strategy avoids this bottleneck by directly mapping discrete ground-truth BEV labels into a learnable embedding space, providing direct supervision without an additional quantization step.
>
> For encoder–decoder architectures, the encoder performs feature extraction only once, while iterative steps occur solely in the decoder. This single shot encoding reduces the potential benefit of iterative processing, and prior work has shown that such iterative steps yield marginal improvements per step [3]. Although incorporating the full encoder–decoder stack into each autoregressive step could address this limitation, it would cause inference cost. Since BEV labels already function as discrete tokens, our class encoding supports a decoder-only design for both feature extraction and iterative steps, which shows improved performance in our experiments.
>
> [Q3] **Inference speed Comparisons**
>
> [A3] We provide the FPS–mIoU trade-off for each autoregressive step in Figure 6 and discuss this behavior in the limitation section. ARINBEV runs at real-time 18 FPS at step 1 while delivering 59.8 mIoU, which is comparable to many prior approaches. This illustrates a practical accuracy speed trade off. Users requiring real-time inference can operate at step 1, while those prioritizing accuracy may choose additional refinement steps. We further compare FPS with a single-shot model (BEVFusion) and iterative methods (DDP and ours).
>
> | Step | BEVFusion | DDP | ARINBEV (ours) |
> |------|-----------|-----|-----------------|
> | 1 | 12 FPS | 10 FPS | 18 FPS |
> | 2 | — | 5 FPS | 9 FPS |
> | 3 | — | 2 FPS | 7.5 FPS |
>
> [Q4] **Feature compression and masking strategies, does not sufficiently discuss how these components interact with the autoregressive core**
>
> [A4] In L315–316, we compress the BEV feature map from 200×200 to 25×25 to reduce the computational cost of deformable cross-attention. The masking strategy enables the model to learn token dependencies by masking ground-truth tokens with a [MASK] token and predicting their categorical distributions. As noted in L89 and L95, learning these masked-token dependencies is important for supporting autoregressive inference. The compression module reduces computation, while the masking strategy directly trains the dependency structure needed for the autoregressive core.
>
> [Q5] **Comparison with BEVFormer**
>
> [A5] Although BEVFormer reports segmentation results, its official repository does not release the corresponding segmentation head and is focused on 3D object detection. However, VQ-Map adopts a similar architecture, leveraging deformable cross-attention to aggregate multi-view camera features, but is supervised through a two-stage VQ-VAE training and does not make use of historical information. Because VQ-Map provides publicly available BEV segmentation code and results, it serves as a more appropriate baseline for comparison with our a single-staged, decoder-only approach.
>
> **References**
>
> [1] Mentzer, Fabian, et al. "Finite Scalar Quantization: VQ-VAE Made Simple." ICLR. 2024.
>
> [2] Yu, Lijun, et al. "Language Model Beats Diffusion-Tokenizer is key to visual generation." ICLR. 2024.
>
> [3] Ji, Yuanfeng, et al. "Ddp: Diffusion model for dense visual prediction." ICCV. 2023.

---

> ### Author Response · Authors · 2025-11-20
> **Rebuttal by Authors**
>
> [Q6] **Different ego trajectory types and performance differences**
>
> [A6] The nuScenes dataset contains multiple scenes, and we subdivide the validation set into straight, left-turn, and right-turn trajectories, as well as moving–stop motion cases. Note that straight scenes contain only straight motion, whereas left-turn and right-turn scenes include straight motion before and after the turning action. Likewise, stop scenes include both stopping moments and brief moving intervals. Because nuScenes was collected at a constant ~30 mph to ensure stable LiDAR capture, the dataset contains limited speed variation. We therefore focus our analysis on trajectory types rather than speed-dependent behavior.
>
> The observed performance differences align with expectations. Turning trajectories generally involve more occlusion from buildings, pedestrians, and surrounding vehicles, which naturally leads to marginally lower mIoU. The left-turn and right-turn trajectories do not differ significantly from each other.
>
> | Trajectory | mIoU |
> |-----------|------|
> | Straight  | 64.5 |
> | Left Turn | 64.0 |
> | Right Turn| 64.1 |
>
> Similarly, the slight drop in performance for the stop case is due to typical stop scenarios occurring at intersections where vehicles stop for traffic lights or to make a turn. These locations have more occlusion, such as crossing pedestrians. This naturally introduces more occlusion and results in slightly lower mIoU compared to continuous motion scenes.
>
> | Motion | mIoU |
> |--------|------|
> | Moving | 64.5 |
> | Stop   | 64.1 |
>
> [Q7] **Generalizability of the Gaussian prior in the entropy-guided masking strategy**
>
> [A7] We appreciate the reviewer for bringing up this important point regarding the generalizability of entropy-guided masking. As shown in Table 4(b), using pure entropy-guided masking produces limited performance due to its central-region bias. To address this issue while still leveraging informative-region masking, we employ a hybrid masking strategy that switches between arccosine-based random masking and entropy-guided masking with probability 0.5 (Table 4(c)). This helps the model to learn informative regions while maintaining supervision across entire BEV grids. A visualization of the arccosine random masking pattern is provided in Figure 8 (bottom) of the Appendix.
>
> Following Reviewer Wj3i’s suggestion, we further evaluated the robustness of entropy-guided masking by dynamically varying the Gaussian prior sigma across several ranges {0.25–0.75}, {0.3–0.6}, and {0.4–0.5}, while keeping the hybrid masking unchanged. These results indicate that the hybrid masking strategy remains stable across different sigma ranges. We will add the results to the appendix and integrate them into the main text in the final version.
>
> | $\sigma$ range  | mIoU |
> |---------------|------|
> | {0.25–0.75} | 63.9 |
> | {0.30–0.60} | 64.2 |
> | {0.40–0.50} | 64.3 |
>
> [Q8] **Declining performance on multi-step inference**
>
> [A8] Beside overfitting, the decline in multi-step inference performance is also observed in other generative [1, 2]. A common explanation is sampling drift, where the distribution during inference differs from training. During training, the model learns to invert a masked ground-truth BEV map, whereas during inference it must iteratively unmask its own imperfect predictions under a fully masked input. As these errors accumulate, the input distribution progressively diverges from the ground-truth distributions, typically after step 3.
>
> To reduce sampling drift, we adopt the same training strategy used in DDP. We construct Z_model from the model’s predictions using unmasked inputs Z, and then apply masking to Z_model during the final 9K iterations (1 epoch). However, we find that accuracy consistently stabilizes around step 3, and additional steps do not yield further improvement. We will include this additional evaluation in the appendix and clarify it in the final version.
>
> | Step | mIoU |
> |------|----------|
> | 1    | 59.8     |
> | 2    | 63.4     |
> | 3    | 64.3     |
> | 4    | 64.3     |
>
> **References**
>
> [1] Ji, Yuanfeng, et al. "Ddp: Diffusion model for dense visual prediction." ICCV. 2023.
>
> [2] Daras, Giannis, et al. "Consistent diffusion models: Mitigating sampling drift by learning to be consistent." NeurIPS. 2023.

---

### Official Review · Reviewer_Qyij · 2025-10-28

**Soundness:** 3
**Presentation:** 3
**Contribution:** 3
**Rating:** 4
**Confidence:** 3

**Summary:**

This paper proposes ARINBEV, a single-stage, decoder-only autoregressive model for BEV map layout prediction. The authors' key contribution is to demonstrate that learning discrete representations in existing two-stage generative models (such as VQ-VAE) is unnecessary given the semantic sparsity of BEV maps. To replace the discrete representations, the authors introduce class encoding and entropy-guided masking based on entropy maps for more efficient and concise training. Experimental results show that ARINBEV achieves new state-of-the-art performance on nuScenes and Argoverse2, while requiring faster training time and fewer model parameters.

**Strengths:**

- Simplicity and Efficiency: The authors have successfully simplified the complex two-stage or encoder-decoder generative framework into a single-stage, decoder-only autoregressive model, significantly reducing model complexity.

- Novel Motivation and Analysis: Through an in-depth analysis of the VQ-VAE codebook utilization and Shannon Entropy of BEV maps, the authors compellingly argue that the semantic sparsity of BEV maps renders discrete representation learning unnecessary.

- Sota Performance: ARINBEV achieves state-of-the-art mIoU performance on both the nuScenes and Argoverse2 datasets, with a training time of 73 hours, which is significantly less than the SOTA baseline (e.g., 131 hours for VQ-Map), while also having fewer parameters, demonstrating an impressive improvement in efficiency.

**Weaknesses:**

- Limited Novelty in Core Architecture: While the removal of VQ-VAE is a reasonable simplification, the core architecture of ARINBEV is essentially based on the decoder-only Transformer from BEVFormer and DETR, and it adopts the autoregressive paradigm of the Masked Generative Image Transformer (MaskGIT). The main innovation—category encoding—essentially maps sparse labels directly to Transformer input embeddings, and its complexity and generalizability still require further validation.

- Generalizability of Entropy-Guided Masking: The "Gaussian Prior Map S" (Equation 10) used for training is a fixed and predetermined prior that focuses on the central area of the BEV. It is merely an empirical, heuristic design rather than a dynamic strategy based on real-time semantic entropy. This undermines the rigor of the "entropy-guided" approach.

**Questions:**

1. Representational Capacity of Category Encoding: The authors convert discrete labels M into embeddings $S_{avg}$ through a learned embedding table E. Please elaborate on how this simple pixel-wise class encoding effectively captures the structural dependencies of traffic elements in the BEV map. How does this "hard-coding" ensure that its expressiveness is sufficiently robust compared to the contextually semantic discrete tokens learned by VQ-VAE?

2. Entropy Guidance and Gaussian Prior: In Section 3.2, the authors mention using an "entropy-guided masking strategy," but actually employ a fixed Gaussian prior map S (Equations 9-10). This Gaussian prior map S simply treats the central area as a high-information region, overlooking the significant differences in traffic scenarios within the BEV coordinate system (e.g., highways vs. complex intersections). The lack of an adaptive masking strategy for complex scenes may limit the model's performance in highly variable environments.

3. Purpose of Using a Fixed Gaussian Prior: As stated in L343, the purpose of employing a fixed Gaussian prior is to prevent the model from overfitting to the entropy distribution of a specific dataset. However, the map prediction network itself is already overfit to a particular dataset. There is no experimental evidence demonstrating that a fixed Gaussian prior is less prone to overfitting compared to an empirical entropy map derived from training data.

4. Decline in Multi-Step Inference Performance: Figure 6 shows that as the number of inference steps increases from 3 to 5, the mIoU declines, while FPS continues to decrease. What is the reason for the drop in mIoU? Is it due to overfitting (as you mentioned), or does the model introduce erroneous information in subsequent iterations?

---

> ### Author Response · Authors · 2025-11-20
> **Rebuttal by Authors**
>
> Dear Reviewer Qyij,
>
> We appreciate the time and attention you devoted to reviewing our submission. We are encouraged by your acknowledgement of **(1) the simplicity and efficiency of our approach, (2) the novelty of the motivation and analysis,** and **(3) the strong performance and training efficiency across datasets.** We are grateful for your constructive feedback and address your concerns below:
>
> [Q1] **Limited novelty in core architecture**
>
> [A1] The novelty of ARINBEV lies not only in its architectural components but also in the overall empirical re-design that simplifies previously complex pipelines for BEV map estimation. Although ARINBEV incorporates elements from BEVFormer/DETR and MaskGIT, we reformulate BEV map estimation as an autoregressive token-prediction problem within a single-stage, decoder-only framework. BEVFormer is a single-shot BEV estimation model, while MaskGIT focuses on autoregressive next-token prediction. Our contribution lies in adapting conditional token-dependency–based autoregression to the BEV setting, enabling iterative BEV refinement. Furthermore, adopting an autoregressive paradigm would typically require a two-stage discrete representation learning pipeline. However, through codebook-utilization and entropy analyses, we show that VQ-VAE–based pipelines can be unnecessary for inherently discrete BEV maps. This empirical finding motivates our class-encoding strategy and the resulting single-stage design.
>
> [Q2] **Representational Capacity of Category Encoding**
>
> [A2] In VQ-VAE–based approaches, the ground-truth BEV map M is encoded into discrete tokens through a quantization layer, and these tokens serve as supervision for the second-stage transformer. Our empirical analysis (Fig. 2–3) shows that these quantized tokens exhibit under-representation, which is consistent with known effects of quantization-induced information loss and can weaken transformer supervision [1, 2]. Because BEV semantic maps are already sparse and inherently discrete, introducing an additional quantization stage forces the model to re-discretize an already discrete space. This creates an unnecessary bottleneck. But, our class-encoding strategy preserves the semantic content by directly mapping discrete ground-truth BEV map M into a learnable embedding space, avoiding quantization entirely and enabling direct supervision.
>
> The robustness of this representation is further supported by Table 3. In Argoverse2 training, two-stage architectures such as MapPrior (−3.9% avg) and VQ-Map (−4.0% avg) show mIoU drops across all classes relative to their nuScenes performance. However, ARINBEV exhibits only a −0.6% average drop, suggesting that class encoding is more stable across datasets than discrete representations learned through VQ-VAE.
>
> | Method   | nuScenes (Drivable) | Argoverse2 (Drivable) | $\Delta$ | nuScenes (Ped. Cross.) | Argoverse2 (Ped. Cross.) | $\Delta$ | nuScenes (Divider) | Argoverse2 (Divider) | $\Delta$ |
> |----------|----------------------|------------------------|----------|--------------------------|----------------------------|----------|----------------------|------------------------|----------|
> | MapPrior | 81.7 | 78.5 | -3.2 | 54.6 | 50.0 | -4.6 | 45.1 | 41.1 | -4.0 |
> | VQ-Map   | 83.8 | 79.7 | -4.1 | 60.9 | 56.9 | -4.0 | 50.8 | 46.9 | -3.9 |
> | Ours     | 85.0 | 83.9 | -1.1 | 62.4 | 61.4 | -1.0 | 51.2 | 51.6 | +0.4 |
>
> [Q3] **Adaptive masking strategy for entropy-guided masking**
>
> [A3] We appreciate the reviewer’s concern about the entropy-guided masking strategy. Table 4(b) shows that using pure entropy-guided masking introduces a central-region bias. To mitigate this issue while benefiting from informative-region masking, we adopt a hybrid masking strategy (Table 4(c)) that switches between entropy-guided masking and arccosine-based random masking with probability 0.5. Combining entropy-guided masking with a random masking scheme helps the model attend to informative regions while maintaining supervision across the spatial grid. An example visualization of the arccosine random masking is provided in Figure 8 (bottom) of the Appendix. We will make sure to clarify this part in the main text.
>
> Following Reviewer Wj3i’s suggestion, we evaluated the adaptability of entropy-guided masking by dynamically varying the $\sigma$ across several ranges {0.25–0.75}, {0.3–0.6}, and {0.4–0.5} during training while keeping the hybrid masking strategy unchanged. These results indicate that the hybrid masking strategy remains stable across different sigma ranges. We will add these results to the appendix and integrate them into the main text in the final version.
>
> | $\sigma$ range   | mIoU |
> |---------------|------|
> | {0.25–0.75}  | 63.9 |
> | {0.30–0.60} | 64.2 |
> | {0.40–0.50} | 64.3 |
>
> **References**
>
> [1] Mentzer, Fabian, et al. "Finite Scalar Quantization: VQ-VAE Made Simple." ICLR. 2024.
>
> [2] Yu, Lijun, et al. "Language Model Beats Diffusion-Tokenizer is key to visual generation." ICLR. 2024.

---

> ### Author Response · Authors · 2025-11-20
> **Rebuttal by Authors**
>
> [Q4] **Purpose of using fixed Gaussian prior**
>
> [A4] Our initial approach was to directly use the smoothed entropy distribution from Figure 2(c) of the nuScenes dataset as a masking probability map. While this yielded a marginal improvement on nuScenes (64.3 > 64.6 mIoU, +0.3%), applying the same entropy distribution to Argoverse2 achieved 65.0 mIoU. But, the masking strategy based on a fixed Gaussian prior resulted in a higher AV2 performance of 65.6 mIoU (+0.6%). These results suggest that the Gaussian prior provides more stable performance across the evaluated datasets and offers a simple approximation of the informative-region trend. For fairness and consistency, we adopt the same hybrid masking strategy with arccosine random masking across all experiments. We will further clarify this in the appendix.
>
> [Q5] **Declining performance on multi-step inference**
>
> [A5] We appreciate the reviewer’s observation regarding the decline in performance across steps. Similar behavior has been reported in other generative models [1, 2]. A common explanation is sampling drift, which occurs when the distributions seen at inference differ from those used during training. During training, the model learns to recover partially masked ground-truth BEV maps, whereas at inference it must iteratively refine its own imperfect predictions starting from a fully masked input. As these imperfections accumulate, the input distribution can diverge from the masked ground-truth distributions and lead to performance degradation beyond step 3.
>
> To reduce this drift, we adopt a training scheme from DDP, where we construct Z_model using the model’s own predictions (input Z without masking) and then apply masking to Z_model during the final 9K iterations (1 epoch). However, despite this mitigation, we observe that performance plateaus after step 3, and additional iteration steps do not yield further improvement. We will include this additional evaluation in the appendix and clarify it in the final version.
>
> | Step | mIoU |
> |------|----------|
> | 1    | 59.8     |
> | 2    | 63.4     |
> | 3    | 64.3     |
> | 4    | 64.3     |
>
> **References**
>
> [1] Ji, Yuanfeng, et al. "Ddp: Diffusion model for dense visual prediction." ICCV. 2023.
>
> [2] Daras, Giannis, et al. "Consistent diffusion models: Mitigating sampling drift by learning to be consistent." NeurIPS. 2023.

---

### Official Review · Reviewer_Wj3i · 2025-10-30

**Soundness:** 3
**Presentation:** 2
**Contribution:** 2
**Rating:** 6
**Confidence:** 3

**Summary:**

This paper presents an empirical analysis showing that current generative BEV map estimation methods contain unnecessary complexity: in particular, discrete representation learning exhibits low codebook utilization and extremely low entropy on BEV maps, leading to feature underrepresentation. Building on this, the authors propose ARINBEV, a conditional autoregressive model with a single-stage, decoder-only design. It directly injects ground-truth BEV map labels into the embedding layer via Class Encoding, thereby avoiding the discrete quantization stage. In addition, the paper introduces an entropy-guided masking schedule that leverages the low-entropy property of BEV maps to focus learning on information-rich regions and capture the conditional dependencies among traffic elements. ARINBEV surpasses existing SOTA methods on nuScenes and Argoverse2, achieving clear improvements.

**Strengths:**

S1. This work brings a novel perspective to BEV layout estimation: through Shannon-entropy analysis and empirical measurements of codebook utilization, it systematically—for the first time—demonstrates the inefficiency of discrete representation learning on BEV maps.
S2. It proposes a single-stage, decoder-only autoregressive BEV generation framework, abandoning the prevalent two-stage VQ-VAE→Transformer and diffusion-based encoder–decoder pipelines; moreover, using Class Encoding, it feeds the labels directly as tokens, a design that is novel and well aligned with the semantic sparsity of BEV maps.
S3. The paper presents the method with clear detail: the autoregressive factorization, embedding normalization, masking schedule, and the compression to 25×25 for global self-attention as well as the cross-attention sampling settings are all documented in the main text and appendix. The appendix also provides PyTorch pseudocode, making the work reproducible.

**Weaknesses:**

W1. The inference procedure relies on iterative autoregressive steps, which leads to higher latency; as noted in the paper, the best case reaches only 7.5 FPS, lagging behind non-generative baselines such as BEVFusion. This setup may constrain deployment in dynamic driving scenarios.
W2. The entropy-guided masking uses a fixed Gaussian prior with p=0.5p=0.5, but the ablation study shows sensitivity to this parameter—for example, the appendix notes that σ=0.3 leads to training failure. This introduces potential instability, and the heavy reliance on a nuScenes-like distribution may degrade on datasets with different entropy patterns.

**Questions:**

Q1. Beyond the best setting at σ=0.5, if the Gaussian prior center is shifted, widened, or narrowed —or if the prior weight pp is varied from 0.5 to {0.25,0.75}—how does the performance trend? Do you observe noticeable fluctuations for classes in the peripheral regions?

---

> ### Author Response · Authors · 2025-11-20
> **Rebuttal by Authors**
>
> Dear Reviewer Wj3i,
>
> We greatly appreciate your thoughtful review and valuable feedback. We are pleased that you recognize **(1) our novel analysis of BEV maps,** **(2) our single-stage, decoder-only autoregressive framework,** and **(3) the clarity and reproducibility of the paper.** We also appreciate your constructive comments and address them as follows:
>
> [Q1] **Inference cost of iterative steps**
>
> [A1] We acknowledge the reviewer’s concern and have discussed this trade-off in the limitation section. ARINBEV already runs at real-time 18 FPS at step 1 while still delivering 59.8 mIoU, which is comparable to many prior approaches. This shows that the iterative design provides a practical accuracy speed trade off: users requiring real-time performance can rely on step 1, while additional steps remain available for improved accuracy when runtime is less constrained.
>
> [Q2-3] **Generalizability of the Gaussian prior in the entropy-guided masking strategy**
>
> [A2–3] We thank the reviewer for raising this concern. We agree that entropy-guided masking may exhibit instability when entropy patterns differ across datasets, and a similar effect is observed in Table 4(b), where pure entropy-guided masking shows central-region bias. To reduce this issue while still leveraging informative-region masking, we adopt a hybrid strategy that alternates between arccosine-based random masking and entropy-guided masking with probability 0.5 (Table 4(c)). This approach helps the model attend to informative central regions while providing more uniform supervision across the BEV grid. A visualization of the arccosine masking pattern is provided in Figure 8 (bottom) of the Appendix. We will clarify this in the main text.
>
> To directly address the reviewer’s question regarding widening or narrowing the Gaussian prior, we conducted additional experiments where sigma $\sigma$ is dynamically varied across three ranges during training {0.25–0.75}, {0.3–0.6}, and {0.4–0.5} while keeping the hybrid masking unchanged. Across all tested ranges, we observed similar mIoU values and did not notice meaningful degradation in peripheral-region classes. These results indicate that the hybrid masking strategy remains consistent performance. Evaluating multiple sigma $\sigma$ values, following the reviewer’s suggestion, provides additional insight into the behavior of the masking strategy. We will add these results to the appendix and summarize them in the main text of the final version.
>
> | $\sigma$ interval  | mIoU |
> |---------------|------|
> | {0.25–0.75}   | 63.9 |
> | {0.30–0.60}   | 64.2 |
> | {0.40–0.50}   | 64.3 |

---

### Meta-Review · Area_Chair_pDxc · 2026-01-05

**Summary:**

The paper proposes ARINBEV, a single-stage, decoder-only autoregressive framework for BEV semantic layout prediction. The main contribution is an empirical finding that discrete representation learning (e.g., VQ-VAE tokenization) is inefficient for sparse, low-entropy BEV maps, motivating a simpler class-encoding formulation and an entropy-guided (hybrid) masking schedule. Reviewers agree the paper is sound, clearly written, and achieves state-of-the-art mIoU on nuScenes and Argoverse2 with improved training efficiency and reduced complexity. The main concerns were (i) novelty relative to prior transformer/autoregressive designs, (ii) robustness/generalizability of the entropy-guided masking (fixed Gaussian prior), and (iii) inference latency due to iterative autoregressive sampling.

AC checked the submission, the reviews, and the rebuttal, and believes that most of the concerns are addressed. Thus, an acceptance is recommended.

**Reviewer Concerns:**

Addressed in rebuttal:
*Masking robustness / Gaussian prior sensitivity: Authors clarify the hybrid masking strategy and provide new experiments varying σ across multiple ranges, showing stable mIoU and no meaningful degradation in peripheral regions, improving confidence in generalizability.
* Purpose of fixed Gaussian prior: Authors provide evidence that Gaussian prior transfers better across datasets than using dataset-specific entropy maps.
* Multi-step inference behavior: Authors explain plateau/limitations via sampling drift, provide additional mitigation discussion, and clarify performance stabilizes around step 3.
* Evaluation clarity (temporal info, inference speed): Authors clarify ARINBEV does not use historical features and provide FPS–mIoU trade-offs (step 1 is real-time), improving practical context.

Still outstanding (but not major blocking):
* Novelty: The core architecture draws on existing components (decoder-only transformer, MaskGIT-style refinement), but the paper’s value lies in the evidence-backed simplification and reformulation for BEV maps.
* Latency at best-accuracy setting: Iterative inference remains slower (e.g., ~7.5 FPS at step 3), but an explicit tradeoff is provided and the limitation is acknowledged.

The rebuttal satisfactorily addresses the key technical concerns (masking robustness, generalizability, inference-step behavior). There are remaining issues (novelty and inference latency tradeoff), but they do not outweigh the overall empirical and practical contribution.

**Reviewer Scores:**

Reviewer Wj3i: likely 6 → 8, masking/generalizability concerns addressed with new experiments; speed/accuracy tradeoff clarified.
Reviewer Qyij: likely 4 → 6, due to better justification and cross-dataset evidence for masking; novelty concerns may keep it near borderline.
Reviewer FeCg: likely 6 → 8, clarified temporal setting, inference overhead, and rationale for why simplification helps.

---

### Decision · Program_Chairs · 2026-01-26

Accept (Poster)